# *Mycobacterium tuberculosis* RpfE-Induced Prostaglandin E2 in Dendritic Cells Induces Th1/Th17 Cell Differentiation

**DOI:** 10.3390/ijms22147535

**Published:** 2021-07-14

**Authors:** Hye-Soo Park, Seunga Choi, Yong-Woo Back, Kang-In Lee, Han-Gyu Choi, Hwa-Jung Kim

**Affiliations:** Department of Microbiology and Department of Medical Science, College of Medicine, Chungnam National University, Daejeon 35015, Korea; 01027192188@hanmail.net (H.-S.P.); seungachoi@cnu.ac.kr (S.C.); lenpk@nate.com (Y.-W.B.); popigletoh@nate.com (K.-I.L.)

**Keywords:** tuberculosis, dendritic cell maturation, PGE2, Th1/Th17 differentiation, naïve T cell differentiation, multifunctional T cell

## Abstract

Prostaglandin E2 (PGE2) is an important biological mediator involved in the defense against *Mycobacterium tuberculosis* (Mtb) infection. Currently, there are no reports on the mycobacterial components that regulate PGE2 production. Previously, we have reported that RpfE-treated dendritic cells (DCs) effectively expanded the Th1 and Th17 cell responses simultaneously; however, the mechanism underlying Th1 and Th17 cell differentiation is unclear. Here, we show that PGE2 produced by RpfE-activated DCs via the MAPK and cyclooxygenase 2 signaling pathways induces Th1 and Th17 cell responses mainly via the EP4 receptor. Furthermore, mice administered intranasally with PGE2 displayed RpfE-induced antigen-specific Th1 and Th17 responses with a significant reduction in bacterial load in the lungs. Furthermore, the addition of optimal PGE2 amount to IL-2-IL-6-IL-23p19-IL-1β was essential for promoting differentiation into Th1/Th17 cells with strong bactericidal activity. These results suggest that RpfE-matured DCs produce PGE2 that induces Th1 and Th17 cell differentiation with potent anti-mycobacterial activity.

## 1. Introduction

Tuberculosis (TB), caused by *Mycobacterium tuberculosis* (Mtb), remains one of the most important infectious diseases worldwide. Mtb infections result in approximately two million deaths annually, indicating an urgent need for improved treatment and prevention strategies [1]. The only vaccine currently available, the *Mycobacterium bovis* BCG (Bacille Calmette-Guérin) vaccine, is both safe and cost-effective and efficiently protects children against early manifestations of TB [2,3]. However, BCG antigens do not persist for a sufficient amount of time to generate long-term effector memory T cells, limiting the efficacy of BCG against adult pulmonary TB [3,4,5]. The reciprocal induction of Th1 and Th17 cell responses plays an important role in establishing protective immune responses against TB [6]. Protection from Mtb is dependent on a robust Th1 response through IFN-γ secretion by antigen-specific CD4^+^ T cells; however, recent studies have emphasized the importance of the Th17 response in protective immunity against Mtb infection [7,8,9]. Hence, understanding the balance between Th1 and Th17 responses during infection and identifying novel proteins that simultaneously induce both Th1 and Th17 responses are crucial in the development of efficacious vaccines.

Eicosanoids, the biologically active lipid mediators derived from arachidonic acid, include prostaglandins, lipoxins, and leukotrienes. In particular, prostaglandin E2 (PGE2) plays an important role in regulating inflammatory responses. PGE2 is a key mediator of pyrexia, hyperalgesia, and arterial dilation, which increases blood flow to inflamed tissues and, in combination with enhanced microvascular permeability, results in edema. The relevance of this pathway in promoting inflammation is supported by the clinical use of cyclooxygenase inhibitors, which are effective anti-inflammatory agents that interfere with prostaglandin synthesis [10]. However, PGE2 can also exert anti-inflammatory properties and negatively regulate the functions of neutrophils, monocytes, and lymphocytes, particularly Th1 cells that produce IFN-γ [11]. This paradox has puzzled many investigators for decades. PGE2 has been shown to exacerbate inflammation and disease severity in murine models of inflammatory bowel disease (IBD) and collagen-induced arthritis through the IL-23–IL-17 pathway [12,13]. These effects have been attributed to the action of PGE2 on innate cells, as PGE2 enhances the production of IL-23 and IL-1β in macrophages and dendritic cells (DCs), while down-regulating IL-12 production [14]. A recent report has shown that PGE2, together with IL-23, favors the expansion of human Th17 cells from peripheral blood mononuclear cells (PBMCs) and that PGE2 enhances IL-17 production induced by IL-23 from memory CD4^+^ T cells [15]. Th17 cells are a unique subset of effector T cells that are distinct from the Th1 and Th2 subsets [16,17,18,19], and they have been implicated as potent effectors of autoimmune disorders, such as multiple sclerosis, psoriasis, arthritis, and IBD [20,21,22,23]. IL-23 and IL-1β are crucial factors in the development of human Th17 cells [22,24,25]. Additionally, the TGF-β- and IL-6-dependent Th17 cell population plays essential roles in chronic inflammation and autoimmunity [26].

Eicosanoids are functionally linked to the immune response that determines the control of Mtb infection in the host. Because the different outcomes of Mtb infection between 5-lipoxygenase (LO)- and cyclooxygenase-2-deficient mice have been reported [27]. Mtb stimulates antigen-presenting cells (APCs) to produce PGE2, which modulates the host immune response through EP2 and EP4 receptors. Mtb infection upregulates EP2 expression in CD4^+^ T cells, and EP2^−/−^ mice have shown higher susceptibility to Mtb infection [28]. IL-1 triggers PGE2 synthesis, type 1 IFNs counter-regulate the prostaglandin axis in Mtb-infected cells, and their networks determine the outcome of TB [29]. However, there are few reports on the effects of PGE2 on T-cell differentiation during Mtb infection and mycobacterial components that induce PGE2 production.

Mtb possesses five rpf homologues, rpfA-E, all of which were expressed in vitro and in vivo [30,31]. Among the genes encoding Rpf family members, *rpfE* was expressed at a higher level during the exponential growth phase under normal culture conditions. Acidic stress and hypoxia also induced the higher relative expression of *rpfE*, indicating that *rpfE* plays a role in cell survival during hypoxia-induced persistence and in the adaptation of Mtb to acid-induced stress conditions [32,33]. During the early stage of Mtb infection of mouse lungs, Mtb are actively growing in mouse organs, thus RpfE expression may be induced at a high level during this stage, making it an important antigen in inducing T cell responses. Despite the importance of Rpf family members in the pathogenesis of TB and in the development of vaccine to prevent Mtb infections, RpfE is less characterized than the other Rpf proteins [34]. We have previously reported that Mtb RpfE-matured DCs simultaneously induce both Th1 and Th17 cell polarization [6]. In this study, we further found that PGE2 produced by RpfE-activated DCs via the MAPK/NF-κB/COX2 signaling pathway induces Th1 and Th17 responses mainly via the EP4 receptor. An optimal dose of PGE2 is essential for promoting differentiation into Th1/Th17 cells with strong bactericidal activity. Our data suggest that RpfE-matured DCs produce PGE2 that leads to Th1 and Th17 cell differentiation with the best anti-mycobacterial activity.

## 2. Results

### 2.1. RpfE Induces PGE2 Production via MAPK/NF-κB Pathway in DCs

Previously, we have reported that RpfE-matured DCs induce both Th1- and Th17-polarized T cell responses [6]. PGE2 has been reported to facilitate the expansion of Th17 [15,35,36] and Th1 [36,37]. To explore whether PGE2 could play a role in RpfE-mediated Th1/Th17 differentiation, we first determined PGE2 production in RpfE-treated DCs. As shown in Figure 1A, RpfE stimulated DCs to secrete PGE2 in a dose-dependent manner. PGE2 production was significantly higher in DCs stimulated with 10 µg/mL RpfE than in those stimulated with 100 ng/mL LPS or untreated cells. Diverse stimuli regulate COX2 induction in immune cells, which mediates PGE2 production from arachidonic acid. DCs stimulated with 10 µg/mL RpfE showed significantly higher COX2 expression compared with untreated cells or LPS-treated cells (Figure 1B). Furthermore, RpfE-mediated PGE2 production was abrogated by pretreatment with an NS398 (COX2 inhibitor) (Figure 1C). Next, we determined that MAPK/NF-κB signaling was involved in RpfE-mediated PGE2 production in DCs. As expected, the RpfE-induced production of IL-1β, IL-12p70 (for Th1 response), and IL-23p19 (for Th17 response) was significantly suppressed in DCs pretreated with pharmacological inhibitors of p38 (SB203580), ERK1/2 (U0126), and NF-κB (Bay 11-7082) but not of JNK (SP600125) (Figure 1D). These inhibitors, except the SP600125 and NS398, inhibited the production of PGE2. However, the NS398 pretreatment did not inhibit the production of RpfE-mediated Th1/Th17-related cytokines. Immunoblot analysis also showed that COX2 expression and NF-κB activation induced by RpfE were inhibited by SB203580, U0126, and Bay 11-7082, but NF-κB expression was not inhibited by NS398 (Figure 1E). These results suggest that MAPK and NF-κB signaling pathways are essential for the production of Th1/Th17-related cytokines and PGE2 as well as for COX2 expression in RpfE-stimulated DCs.

### 2.2. PGE2 Produced by RpfE-Matured DCs Induces Th1 and Th17 Differentiation Mainly via EP4 Receptor

Next, we investigated whether cytokines and PGE2 produced by RpfE-stimulated DCs could affect Th1 and Th17 differentiation. DCs pretreated with a pharmacological inhibitor were matured with RpfE and then co-cultured with splenocytes from Mtb-infected mice. The NS398 and all pharmacological inhibitors except the SP600125 significantly suppressed the production of IFN-γ and IL-17A cytokines (Figure 2A) and the expression of T-bet-inducing Th1 response and RoRγt-inducing Th17 response (Figure 2B). PGE2 induces Th1 and/or Th17 differentiation through EP2 and EP4 receptors [35,36]. Therefore, we determined whether EP receptors are involved in RpfE-induced Th1 and Th2 responses. Under the same conditions as shown in Figure 2A, the production of RpfE-mediated IFN-γ and IL-17A was significantly suppressed by pretreatment with EP2 or EP4 antibody before co-culturing with splenocytes. The expression of IFN-γ and IL-17A was significantly inhibited by EP4 antibody than EP2 antibody compared to untreated conditions (Figure 2C). Simultaneous treatment with the two antibodies significantly inhibited cytokine production compared to the single antibody treatment. RpfE-mediated T-bet expression was also significantly inhibited by EP4 antibody and slightly inhibited by EP2 antibody, but their inhibition was enhanced by co-treatment with both antibodies (Figure 2D). Expression of RoRγt was prominently inhibited by the EP4 antibody but not by the EP2 antibody. Next, to confirm that these differentiated T cells play a role in controlling Mtb infection, the antibacterial activity of the T cells activated by RpfE-treated DCs was determined under conditions with or without pretreatment with EP2/4 antibodies. When T cells purified from splenocytes activated with RpfE-matured DCs were co-cultured with Mtb-infected macrophages, the growth of intracellular Mtb was significantly inhibited compared to that in non-activated T cells (Figure 2E). The T cells treated with EP4 antibody before co-culturing with Mtb-infected macrophages did not show significant inhibition of Mtb growth. Although EP2 antibody-treated T cells still showed significant Mtb growth inhibition, their anti-bactericidal activity was lower than that of untreated T cells. The inhibition of RpfE-mediated Mtb growth was completely abolished by neutralization of EP4 and EP2 with both the antibodies. Taken together, these results suggest that PGE2 produced by RpfE-stimulated DCs induces Th1 and Th17 responses mainly via the EP4 receptor and partly via the EP2 receptor, which inhibits intracellular Mtb growth.

### 2.3. Induction of RpfE-Specific Th1 and Th17 Responses Is Dependent on PGE2 In Vivo

To investigate whether RpfE could induce Th1 and Th2 responses in vivo in a PGE2-dependent manner, which could also contribute to Mtb control, Mtb-infected mice were intranasally administrated six times at two-day intervals with RpfE without an adjuvant. Four weeks after the last treatment, RpfE administration induced a significant reduction in bacterial load in the lungs (Figure 3A). However, bacterial loads in mice injected with NS398 before the first RpfE administration were significantly restored compared to those of the RpfE only-administrated group but were still significantly lower compared to those of the infection control group. Changes in the T-cell phenotypes in mice administrated with RpfE were analyzed using fluorescence-activated cell sorting (FACS). As shown in Figure 3B, the population of IFN-γ^+^- and IL-17^+^-CD4^+^ T cells in the lungs of RpfE-administrated mice, but not in NS398-injected mice, was increased after re-stimulation of lung cells with RpfE. In addition, the number of IL-17^+^ T cells, but not IFN-γ^+^ T cells, was reduced by NS398 injection (Figure 3B). Cytokines and PGE2 production from the culture supernatants under the same conditions as those in Figure 3B were also determined. As shown in Figure 3C, the production of all cytokines tested and PGE2 were significantly higher in re-stimulated lung cells of the administrated mice with RpfE than in the non-treated mice. However, the production of IL-1β, IL-12p70, IL-23p19, IFN-γ, IL-17A, and PGE2 was significantly suppressed in mice pre-injected with NS398; in particular, the production of IL-23p19, IL-17A, and PGE2 was almost abolished by NS398 injection (Figure 3C). These data suggest that RpfE-induced Th1 and Th17 responses are partially PGE2-dependent, and the Th17 response is more significantly PGE2-dependent than the Th1 response.

### 2.4. Optimal Dose of PGE2 Is Essential to Promote Th1/Th17 Differentiation with Strong Bactericidal Activity

Next, we specifically investigated the PGE2-mediated effect on Th1/Th17 development through networking with other cytokines and the anti-mycobacterial activity of these differentiated T cells. Naïve CD4^+^ T cells were cultured in the presence of IL-2 and cytokines related to the differentiation of T cells and then co-cultured with Mtb-infected macrophages. For example, IL-6 and TGF-β were selected for Th17 differentiation and IL-23p19 for maintenance of the Th17 response [38]. As expected, the expression of GATA-3 and T-bet was increased in T cells cultured with IL-2-IL-4 and IL-2-IL-12p70 (Figure 4A), and IL-2-IL-12-cultured T cells, but not IL-2-IL-4-cultured T cells, significantly inhibited Mtb growth in macrophages compared to the infection control (Figure 4B). T cells cultured with IL-2-IL-6-TGF-β or IL-2-IL-6-IL-23p19-TGF-β expressed only RoRγt (Figure 4A) and did not inhibit Mtb growth (Figure 4B). In contrast, T cells differentiated with IL-2-IL-6-IL-23p19-IL-1β expressed both T-bet and RoRγt and significantly inhibited Mtb growth compared to the infection control. The expression of both transcription factors and bacterial growth inhibition were significantly enhanced by PGE2 addition under these conditions (i.e., IL-2-IL-6-IL-23p19-IL-1β-PGE2) (Figure 4A,B). The IFN-γ, IL-17A, and IL-5 levels in the supernatants from the same culture conditions as those in Figure 4B were determined. The significant production of IFN-γ and IL-17A coincided with culture conditions with T-bet and/or RoRγt expression (Figure 4C), and significant IL-5 production was observed only in GATA-3-expressed conditions (i.e., IL-2-IL-4-cultured T cells).

PGE2 exerts both immunosuppressive and pro-inflammatory roles in T-cell responses in a PGE2 concentration-dependent manner [39,40]. Therefore, we determined the antibacterial activity of these T cells differentiated with IL-2, IL-6, IL-23p19, IL-1β, and PGE2 (1–100 ng/mL) in Mtb-infected macrophages. T cells differentiated in the presence of 10 ng/mL PGE2 significantly inhibited Mtb growth compared to the infection control, and their bactericidal activity was comparable with that of T cells activated by RpfE-matured DCs (Figure 4D). In contrast, T cells activated with 1 ng/mL PGE2 did not significantly inhibit Mtb growth compared to the infection control, and T cells developed with a high concentration of PGE2 (>50 ng/mL) did not induce a bactericidal activity in macrophages. Under the same conditions as in Figure 4D, significant IFN-γ was produced in the presence of 10 ng/mL PGE2 but not at higher concentrations; in contrast, IL-17A was produced in a PGE2 dose-dependent manner (Figure 4E). In fact, COX2 expression was significantly induced in macrophages with avirulent Mtb H37Ra but not Mtb H37Rv, and the addition of PGE2 (10–100 ng/mL) in Mtb-infected macrophages did not inhibit bacterial growth (Appendix A). These data suggest that IFN-γ and optimal IL-17 are required for T cell differentiation with strong bactericidal activity, and differentiation is induced in these T cells by an optimal concentration of PGE2. Taken together, these results indicate that RpfE-matured DCs produce PGE2 that leads to Th1 and Th17 development with the best anti-mycobacterial activity.

## 3. Discussion

A previous study revealed that RpfE-matured DCs stimulate naïve T cells and antigen-specific T cells to produce IFN-γ and IL-17 [6]. However, the underlying mechanism inducing the Th1- and Th17-immune responses remains unclear. Here, we report that PGE2 is produced by RpfE-activated DCs via the MAPK/NF-κB/COX2 signaling pathway, resulting in Th1/Th17 cell differentiation with mycobactericidal activity mainly via the EP4 receptor.

To elicit Th1-cell-based immunity is based on evidence from various animal models that a strong IFN-γ-mediated Th1 immune response is the primary protective mechanism of anti-TB immunity [41,42,43,44,45,46]. However, an IFN-γ response is not an optimal correlate of protection [47,48,49] because IFN-γ alone is insufficient to control Mtb infection [50]. Previous studies have demonstrated that antigens that do not elicit Th1 responses uniformly fail to protect against Mtb, but not all proteins that induce robust Th1 responses after vaccination provide considerable protection [46,51,52,53]. In addition, Th17 cells also contribute to protective TB immunity in mice [54,55,56], cynomolgus macaques [57], and rhesus macaques [58]. Although the role of Th17 cells in human patients appears ambiguous [59,60], early Th17 responses are suppressed in Mtb-infected patients as compared to healthy controls [61]. Therefore, the identification of a mechanism inducing RpfE-mediated Th1 and Th17 responses is important to understand subunit vaccine-induced protective responses.

PGE2 regulates a multitude of functions in cell activation/suppression, differentiation, maturation, homeostatic, and inflammatory responses. Focusing on the relationship between PGE2 and T lymphocytes, in the past, PGE2 produced by APCs inhibited the production of IL-2 and IFN-γ and suppressed the proliferation of murine and human T cells in vitro [62,63]. However, a previous study demonstrated that exogenously added low-dose PGE2 potentiates Th1 and Th17 responses and aids in T-cell proliferation [14,35,36], which are related to the underlying mechanisms of various immune diseases. Virulent Mtb inhibits COX2 expression in macrophages, resulting in inhibition of the PGE2-inducing apoptotic response and prevention of necrotic response [64]. In this study, we also found that avirulent Mtb H37Ra, but not H37Rv strain, induced significant COX2 expression in DCs. However, there are no reports on the mycobacterial components regulating PGE2 or COX2 production in APCs. Interestingly, our data suggest that RpfE-activated DCs produce PGE2 via COX2 expression, indicating that the NS398 suppressed the RpfE-mediated PGE2 production. The NF-κB and MAPK pathways have been reported to induce COX2 expression in activated macrophages [65]. Furthermore, activated NF-κB translocates into the nucleus and then secretes Th1/Th17-related cytokines, such as IL-1β, IL-6, IL-12p70, and IL-23p19 [6]. Therefore, our data support that RpfE-activated DCs produce PGE2 through the MAPK/NF-κB/COX2 signaling pathway and that PGE2 may be a key factor inducing the Th1/Th17 response (Figure 1, Figure 2 and Figure 3).

The heterogeneous effects of PGE2 are reflected by the existence of four different PGE2 receptors, designated EP1, EP2, EP3, and EP4, among which EP2 represents low-affinity receptors, whereas EP4 requires significantly lower concentrations of PGE2 for effective signaling [40]. Therefore, EP2 and EP4 signaling is triggered by different concentrations of PGE2, which differs in duration. EP4 signaling is rapidly desensitized following its interaction with PGE2, whereas EP2 is resistant to ligand-induced desensitization, indicating its ability to mediate PGE2 functions over prolonged periods of time and at later time points of inflammation [66]. Kaul et al. reported that EP2-deficient mice showed increased susceptibility to Mtb infection [28]. Previous studies have demonstrated that low-dose PGE2 potentiates Th1 and Th17 responses of mouse T cells in vitro through the EP2 and EP4 receptors [39,67]. In the present study, our results support that PGE2 produced by RpfE-stimulated DCs induces Th1 and Th17 responses mainly via the EP4 receptor and minor EP2 receptor. Blocking EP4 on T cells led to a more profound inhibition of IFN-γ and IL-17 production and abrogation of RpfE-mediated Mtb growth inhibition (Figure 2). Furthermore, we also demonstrated that mice administered nasally with RpfE showed a significant reduction in bacterial load in the lung, and these effects of RpfE were partially lost by injection of NS398 before RpfE administration (Figure 3).

Considering the substantial contributions of Th differentiation in the inhibition of Mtb growth, it is important to reveal the cytokine networks that regulate Th differentiation from naïve CD4^+^ T cells. To do this, it is essential to establish a Th differential system using various related cytokines. Therefore, we conducted an experiment using an in vitro naïve T-cell differentiation system to confirm whether PGE2 affects T-cell differentiation (Figure 4). Th1 polarization is primarily driven by IL-12 and IFN-γ, while Th2 polarization is primarily driven by IL-4. These cytokine signals via STAT4, STAT1, and STAT6 directly control the transcription factors T-bet and GATA3, which, in turn, determine Th1 and Th2 differentiation, respectively [68]. Th1 cells produce IFN-γ, which facilitates their differentiation while inhibiting IL-4-mediated Th2 differentiation [69]. Reciprocally, Th2 cells produce IL-4 and IL-10, which strongly inhibit IL-12/IFN-γ-driven Th1 differentiation. In the present study, we confirmed that T cells cultured with IL-2-IL-12 and IL-2-IL-4 differentiated into Th1 and Th2 cells (Figure 4). The Th17 differentiation of naïve T cells is initiated by IL-6 and TGF-β [70,71,72,73]. In addition, IL-23 and IL-21 are thought to be key cytokines involved in the maturation and/or maintenance of Th17 cells [74,75,76]. IL-6, IL-21, and IL-23 activate STAT3, which is essential for Th17 differentiation. STAT3 plays a critical role in the induction of the orphan nuclear receptor RoRγt, which directs Th17 cell differentiation by inducing the IL-23 receptor. Our data suggest that T cells cultured in the presence of IL-2-IL-6-TGF-β differentiate into Th17 cells, but not Th1 cells, irrespective of the presence of IL-23. Although TGF-β is clearly required for the generation of Th17 cells, TGF-β indirectly favors the expansion of Th17 cells through the inhibition of Th1 cell development [77]. In contrast, CD4^+^ T cells that develop in the presence of IL-1β can develop into IL-17-and IFN-γ-secreting cells [78]. In this study, T cells in the presence of IL-2, IL-6, and TGF-β with or without IL-23p19 expressed the Th17 transcription factor but not the Th1 transcription factor. However, both the transcription factors for Th1 and Th17 differentiation were expressed when IL-1β, but not TGF-β, was added under these culture conditions (i.e., IL-2-IL-6-IL-23p19-IL-1β), and both IFN-γ and IL-17 were produced by these T cells. The T cells differentiated in the presence of PGE2 (i.e., IL-2-IL-6-IL-23p19-IL-1β-PGE2) produced significantly higher levels of IFN-γ and IL-17 than those in the absence of PGE2 (Figure 4C) and showed the most prominent inhibition of Mtb growth in macrophages (Figure 4B). Furthermore, the optimal concentration of PGE2 in differentiated T cells was critical to elicit the best bactericidal effects (Figure 4D). In a previous study, high concentrations of PGE2 favored IL-17 production and down-modulation of IFN-γ production [79], but low concentrations of PGE2 potentiated Th1 and Th17 differentiation [36]. Therefore, it is suggested that PGE2 induces Th1/Th17 cell differentiation at an optimal concentration, which enhances the killing effect on Mtb.

In conclusion, our study shed light on a paradox in which the important role of PGE2 in mediating inflammation remains controversial with respect to its effects on T helper cell function and cytokine production. Our findings also highlights that RpfE-activated DCs produce an optimal amount of PGE2 to induce the differentiation of naïve T cells into Th1/Th17 cells with strong bactericidal activity. In a recent study, the importance of Th1 and Th17 in the effective control of Mtb has been emphasized in mice and humans [80,81]. In addition, our previous study also published the highest correlation between IFN-γ/IL-17 co-producing CD4^+^ T cells and Mtb growth inhibition [46]. Therefore, the results provide that RpfE, which can induce Th1/Th17 by producing the optimal dose of PGE2, has potential Mtb vaccine candidate.

## 4. Materials and Methods

### 4.1. Animals

Specific pathogen-free female C57BL/6 (H-2K^b^ and I-A^b^) mice were purchased from the Jackson Laboratory (Bar Harbor, ME, USA) at 5–6 weeks of age and were maintained under barrier conditions in a biohazard animal room at the Medical Research Center of Chungnam National University, Daejeon, Korea. The animals were fed a sterile commercial mouse diet and were provided water Ad libitum. All animal experiments complied with the ethical and experimental regulations for animal care of Chungnam National University (CNU-00284).

### 4.2. Cell Preparation

Murine bone marrow-derived DCs (BMDCs) were generated, cultured, and purified as described previously [82]. Bone marrow-derived macrophages (BMDMs) were prepared using recombinant M-CSF (CreaGene, Gyeonggi, Korea), as previously described [82]. Briefly, bone marrow cells isolated from C57BL/6 mice were lysed with red blood cell (RBC)-lysing buffer (ammonium chloride 4.15 g/500 mL, 0.01 M Tris-HCl buffer pH [7.5 ± 0.2]) and washed with RPMI 1640 medium. The obtained cells were plated in 6-well culture plates (10^6^ cells/mL, 3 mL/well) and cultured at 37 °C in the presence of 5% CO_2_ in RPMI 1640 media supplemented with 100 unit/mL penicillin/streptomycin (Lonza, Basel, Switzerland), 10% fetal bovine serum (Lonza, Basel, Switzerland), 50 μM mercaptoethanol (Lonza, Basel, Switzerland), 0.1 mM non-essential amino acids (Lonza, Basel, Switzerland), 1 mM sodium pyruvate (Sigma, St. Louis, MO, USA), 20 ng/mL GM-CSF (CreaGene, Gyeonggi, Korea), and 10 ng/mL IL-4 (for BMDCs) or 20 ng/mL M-CSF (for BMDMs).

Age- and sex-matched C57BL/6 mice were infected with Mtb H37Ra ATCC 25177. Briefly, following anesthetization using a xylazine:zoletil (9:1) mixture, 12 mice per group were intratracheally infected with 50 µL of suspensions to achieve initial infectious doses of 10,000 CFU of H37Ra per mouse lung. Five to six mice were sacrificed at eight weeks post-infection, and their spleens were isolated under sterile conditions. Then, the spleen cells and aggregates were filtered through a 40 μm cell strainer in Dulbecco’s phosphate-buffered saline (PBS) using a sterile 1 mL syringe. The erythrocytes were lysed using RBC lysis buffer (90 mL 0.16 M NH_4_ and 10 mL of 0.17 M Tris [pH 7.65]) for 2 min at room temperature. Total spleen (1 × 10^7^ cells/well) from H37Ra-infected mice were stimulated using RpfE (10 μg/mL).

Mtb-infected mice were vaccinated six times at two-day intervals with RpfE (20 μg) by intranasal injection following anesthesia with an intraperitoneal injection of xylazine:zoletil (9:1). In the NS398 (Abcam, Cambridge, UK) treated group, RpfE was added at the time of initial administration. Four weeks after the last vaccination, the mice were euthanized. The lungs were isolated under sterile conditions. These organs were cut into 0.5 cm pieces and agitated in 5 mL of cellular dissociation buffer (RPMI medium containing 0.1% collagenase type IV (Worthington Biochemical Corporation, Lakewood, NJ, USA), 1 mM CaCl_2_, and 1 mM MgCl_2_ for 15 min at 37 °C. Then, the lung cells and aggregates were filtered through a 40 μm cell strainer in Dulbecco’s PBS using a sterile 1 mL syringe. Erythrocytes were lysed using RBC lysis buffer for 2 min at room temperature. Total lung cells (1 × 10^7^ cells/well) from RpfE-vaccinated mice were stimulated using RpfE (10 μg/mL).

CD4^+^ T cells were isolated from the spleens of non-infected mice using a MACS LS CD4-positive selection column. Purified CD4^+^ T cells (5 × 10^6^ cells/well) were co-cultured with DC subsets (5 × 10^5^ cells/well) and stimulated with RpfE. Cells were collected at 48 h for western blot analysis of T-bet, RoRγt, and GATA-3 levels. The supernatants were collected at 72 h, and the cytokine levels were determined using ELISA.

### 4.3. Bacterial Counts

Adherent BMDMs (2 × 10^4^ cells/well) were washed twice in PBS and infected in triplicate with Mtb (2 × 10^4^ bacilli/well). Tubercle bacilli and macrophages were incubated for 4 h. Then, the infected BMDMs were treated with amikacin (200 μg/mL) for 2 h. After 2 h, the monolayers were washed to remove extracellular bacilli. Two different types of CFU measurements in infected macrophages were performed. First, a previously prepared mixture was added to each well, and the plate was incubated for 3 days. The mixture consisted of antigen-activated DCs co-cultured with CD4^+^ sorted T cells at a DC:T cell ratio of 1:10 for 3 days. These CD4^+^ T cells were pre-incubated with neutralizing antibodies against EP2 (R&D Systems, Minneapolis, MN, USA) (200 ng/mL), EP4 (R&D Systems) (200 ng/mL), or EP2/EP4 (200 ng/mL) for 2 h at a DC:T cell ratio of 1:10 for 3 days. The DC-activating antigens were RpfE (10 μg/mL). Second, the infected BMDMs were co-cultured with differentiated T cells incubated with cytokines. All cytokine used were purchased from R&D Systems. The differentiated T cells were naïve CD4^+^ T cells (1 × 10^6^) cultured with each cytokine (5 ng/mL IL-2, 25 ng/mL IL-4, 5 ng/mL IL-12, 25 ng/mL IL-6, 250 ng/mL TGF-β, 50 ng/mL IL-23, 10 ng/mL IL-1β, and 5 ng/mL PGE2) in 24-well culture plates coated with anti-CD3 antibodies (2 μg/mL) for 5 days. The number of ingested and internalized Mtb by BMDMs was calculated by lysing the infected cells from one of the wells in distilled water. The Tubercle bacilli counts of the inoculum were then checked by serial dilution and plating on 7H10 agar with 10% Middlebrook OADC supplement (Difco, Detroit, MI, USA). The plates were incubated at 37 °C for three weeks. At the end of the three-week period, plates were taken out, and colony-forming units (CFUs) were calculated from the number of Mtb colonies.

Four weeks after the last immunization, the lungs were dissected from the infected mice and homogenized. The number of viable bacteria was determined by plating serial dilutions of the lung homogenates onto Middlebrook 7H10 agar (Difco Laboratories, Detroit, MI, USA) supplemented with 10% OADC (Difco Laboratories) and amphotericin B (Sigma-Aldrich, St. Louis, MO, USA). Colonies were counted after four weeks of incubation at 37 °C.

### 4.4. Intracellular Cytokine Staining

Four weeks after the last immunization, the mice were euthanized by CO_2_ asphyxiation, and single-cell suspensions (1 × 10^6^ cells) from immunized mice were re-stimulated with RpfE (10 μg/mL) at 37 °C for 12 h in the presence of both GolgiPlug and GolgiStop (BD Biosciences, Franklin Lakes, NJ, USA). PBS-washed cells were blocked with anti-CD16/32 antibody (BD Biosciences) at 4 °C for 20 min. The cells were then surface stained with Brilliant Violet (BV) 605-conjugated anti-CD4 antibody (BD Biosciences) at 4 °C for 30 min and washed three times with PBS. These cells were fixed and permeabilized with the Cytofix/Cytoperm kit (BD Biosciences) at 4 °C for 30 min following the manufacturer’s instructions. Cells were then washed three times with Perm/Wash (BD Biosciences) and stained intracellularly with PE-conjugated anti-IFN-γ (BD Biosciences) and FITC-conjugated anti-IL-17 (BD Biosciences) at 4 °C for 30 min. After washing three times with Perm/Wash, the cells were fixed using IC fixation buffer (eBioscience, San Diego, CA, USA). Intracellular cytokine levels were detected using the software program Novocyte (Acea Biosciences, San Diego, CA, USA) and analyzed using the software program FlowJo (Treestar, Inc., San Carlos, CA, USA).

### 4.5. Cytokine Measurement

Sandwich enzyme-linked immunosorbent assay (ELISA) was used to determine the levels of PGE2, IL-1β, TNF-α, IFN-γ, IL-17A, IL-23p19, IL-12p70, and IL-10 in the culture supernatants, as previously described [83]. These cytokine assays were performed as recommended by the antibody manufacturers (eBioscience and BD Biosciences).

### 4.6. Immunoblotting Analysis

Stimulated DCs or differentiated T cells were lysed in 100 µL of lysis buffer containing 50 mM Tris-HCl (pH 7.5), 150 mM NaCl, 1% Triton-X100, 1 mM EDTA, 50 mM NaF, 30 mM Na_4_PO_7_, 1 mM phenylmethanesulfonyl fluoride, 2 µg/mL aprotinin, and 1 mM pervanadate. Immunoblotting was performed as previously described [83]. Epitopes of the target proteins, including COX2, NF-κB, T-bet, and RoRγt, were labeled using specific antibodies, and the bands were visualized using the ECL Advance Kit (GE Healthcare, Little Chalfont, UK).

### 4.7. Nuclear Extract Preparation

Nuclear extracts were prepared by first treating stimulated DCs or differentiated T cells with 100 µL of lysis buffer (10 mM HEPES [pH 7.9], 10 mM KCl, 0.1 mM EDTA, 0.5% Nonidet P-40, 1 mM dithiothreitol, and 0.5 mM PMSF) on ice for 10 min. The lysates were centrifuged at 4000 rpm for 5 min, and the pellet was resuspended in 100 µL of extraction buffer (20 mM HEPES [pH 7.9], 400 mM NaCl, 1 mM EDTA, 1 mM DTT, and 1 mM PMSF) and incubated on ice for 30 min. Following centrifugation at 12,000 rpm for 10 min, the supernatant containing the nuclear extracts was collected and stored at −80 °C until use.

### 4.8. Treatment of DCs with Pharmacological Inhibitors of Signaling Pathways

All pharmacological inhibitors used were purchased from Calbiochem (San Diego, CA, USA). Dimethyl sulfoxide (Sigma) was added to the cultures to a final concentration of 0.1% (*v*/*v*) as solvent control. DCs were washed using PBS and pretreated with inhibitors in RPMI 1640 medium containing glutamine for 1 h prior to treatment with RpfE for 24 h. The inhibitors were used at the following concentrations as determined by careful titration: U0126 (10 µM), SB203580 (20 µM), SP600125 (10 µM), Bay11-7082 (20 µM), and NS398 (10 µM). The viability of the DCs was assessed using an MTT assay.

### 4.9. Statistical Analysis

All experiments were repeated at least three times, with consistent results. Significant differences between two groups were determined by unpaired Student’s *t*-test, or those among more than three groups were evaluated with one-way ANOVA followed by Tukey’s multiple comparison test using GraphPad Prism Software, version 4.03 (GraphPad Software Inc., San Diego, CA, USA). Data are expressed as the mean ± standard deviation (SD). Values with * *p* < 0.05, ** *p* < 0.01, and *** *p* < 0.001 were considered statistically significant.

## Figures and Tables

**Figure 1 ijms-22-07535-f001:**
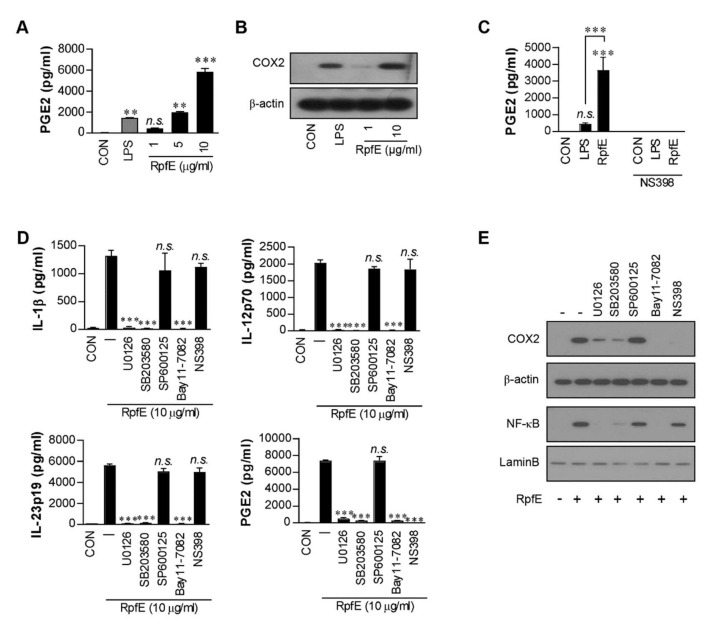
*Mycobacterium tuberculosis* (Mtb) RpfE produces prostaglandin E2 (PGE2) in activated dendritic cells (DCs) via the MAPK/NF-κB/COX2 signaling pathway. (**A**) DCs (1 × 10^6^ cells/mL) were cultured in the presence of 1, 5, or 10 µg/mL RpfE or 100 ng/mL LPS for 24 h, and the levels of PGE2 in the culture supernatant were quantified using ELISA. Data are the mean ± SD from three independent experiments; ** *p* < 0.01 and *** *p* < 0.001 compared to untreated DCs. *n.s.*: no significant difference. (**B**) DCs were treated with 1 or 10 µg/mL RpfE or 100 ng/mL LPS for 4 h. Cell lysates were subjected to SDS-PAGE, and immunoblotting analysis was performed using COX2 antibody. β-actin was used as the loading control for the cytosolic fraction. (**C**) DCs were treated with pharmacological inhibitors of NS398 (COX2 inhibitor, 10 μM) for 1 h prior to treatment with 10 µg/mL RpfE or 100 ng/mL LPS for 24 h. The levels of PGE2 in the culture media were determined using ELISA. Data are the mean ± SD from three independent experiments; *** *p* < 0.001 compared to untreated DCs. *n.s.*: no significant difference. (**D**) DCs were treated with pharmacological inhibitors of p38 (SB203580, 20 µM), ERK1/2 (U0126, 10 µM), JNK (SP600125, 20 µM), NF-κB (Bay11-7082, 20 µM), and NS398 (10 μM) for 1 h prior to treatment with 10 µg/mL RpfE for 24 h. The levels of IL-1β, IL-12p70, IL-23p19, and PGE2 in the culture media were determined using ELISA. Data are the mean ± SD from three independent experiments; *** *p* < 0.001 compared to RpfE treated DCs. *n.s.*: no significant difference. (**E**) DCs were treated with pharmacological inhibitors for 1 h prior to treatment with 10 µg/mL RpfE for 4 h. Cell lysates were subjected to SDS-PAGE, and immunoblotting analysis was performed using COX2 or NF-κB antibodies. β-actin and Lamin B were used as the loading controls for the cytosolic and nuclear fractions, respectively.

**Figure 2 ijms-22-07535-f002:**
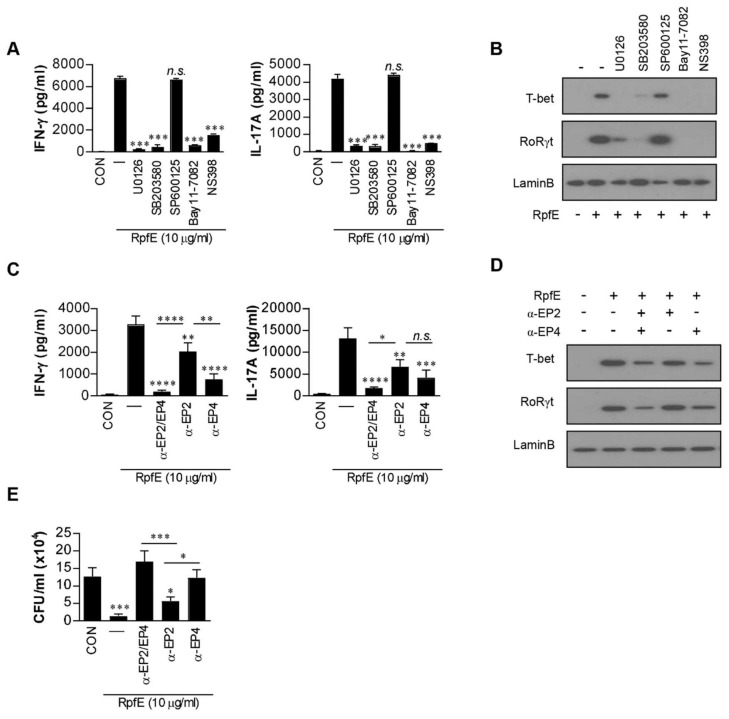
Prostaglandin E2 (PGE2) produced by RpfE-activated dendritic cells (DCs) induces Th1/Th17 differentiation via the EP4 receptor. (**A**,**B**) CD4^+^ T cells were isolated using MACS from Mtb-infected mice splenocytes and co-cultured for 72 h with dendritic cells (DCs) treated with pharmacological inhibitors for 1 h prior to treatment with 10 µg/mL RpfE. (**A**) IFN-γ or IL-17A production in T cells co-cultured with RpfE-treated DCs was determined using ELISA in triplicate (*n* = 3). All data are expressed as the mean ± SD; *** *p* < 0.001 for treated samples compared to RpfE-treated DCs co-cultured with CD4^+^ T cells; *n.s.*, no significant difference. (**B**) The levels of T-bet and RoRγt in T cells were assessed by immunoblotting using specific anti-T-bet and anti-RoRγt monoclonal antibodies. (**C**,**D**) T cells in the presence or absence of neutralizing antibody (anti-EP2, anti-EP4, or anti-EP2/EP4) co-cultured with unstimulated DCs or RpfE-stimulated DCs for 3 days. (**C**) The cytokine levels in culture supernatants were determined using ELISA. (**D**) The cell lysates were analyzed by immunoblotting. “α” means neutralization antibody treatment. All data are expressed as the mean ± SD; * *p* < 0.05, ** *p* < 0.01, *** *p* < 0.001 and **** *p* < 0.0001 for treated samples compared to RpfE-treated DCs co-cultured with CD4^+^ T cells; *n.s.*, no significant difference. (**E**) T cells in the presence or absence of neutralizing antibody (anti-EP2, anti-EP4, or anti-EP2/EP4) activated with unstimulated DCs or RpfE-stimulated DCs at a DC:T cell ratio of 1:10 for 3 days were co-cultured with Mtb-infected bone marrow-derived macrophages (BMDMs). Intracellular Mtb growth in the BMDMs was determined after 3 days. All data are expressed as the mean ± SD; * *p* < 0.05 and *** *p* < 0.001 for treated samples compared to infected BMDM co-cultured with activated CD4^+^ T cells by RpfE-treated DCs; *n.s.*, no significant difference.

**Figure 3 ijms-22-07535-f003:**
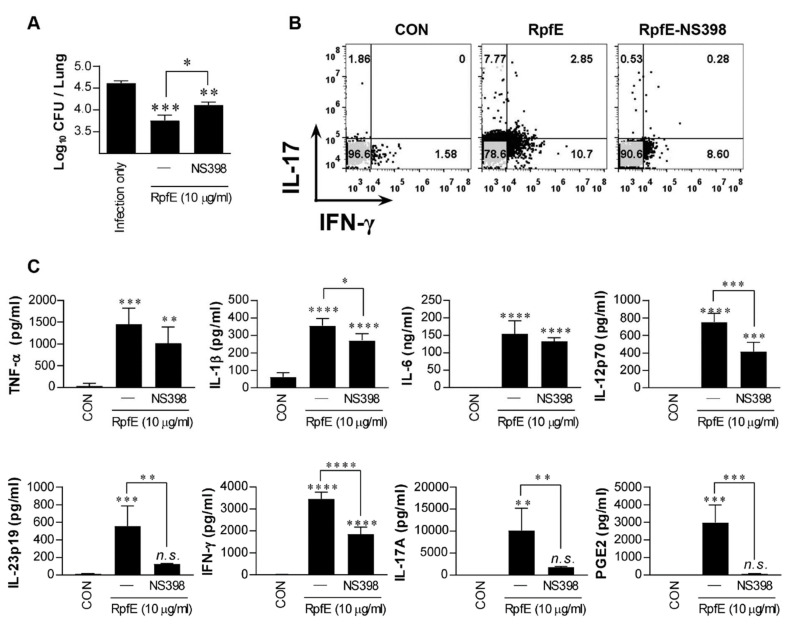
Prostaglandin E2 (PGE2)-dependent induction of RpfE-specific Th1 and Th17 immune responses in vivo. Mtb-infected mice were administrated six times at two-day intervals with RpfE (20 μg) by intranasal injection. In the NS398 treated group, RpfE was added at the time of initial administration. (**A**) Four weeks after the last administration, the lung bacillary burden (log10 CFU) was determined. Data are expressed as the mean ± SD for 5 mice from each group. *n.s.*: not significant, * *p* < 0.05, ** *p* <0.01 and *** *p* < 0.001 compared to infection control mice. *n.s.*: no significant difference, * *p* < 0.05 between RpfE- and RpfE+NS398-treated mice. (**B**) Lung cells were stimulated with RpfE (10 μg/mL), and antigen-specific CD4^+^ T cells were evaluated for IFN-γ and IL-17A secretion by intracellular cytokine staining. (**C**) Levels of cytokine secreted by lung cells in response to RpfE (10 μg/mL) stimulation as detected by ELISA in all treatment groups. Data are expressed as the mean ± SD for 5 mice from each group. *n.s.*: no significant difference, ** *p* < 0.01, *** *p* < 0.001 and **** *p* < 0.0001 compared to infection control mice. *n.s.*: no significant difference, * *p* < 0.05, ** *p* < 0.01, *** *p* < 0.001 and **** *p* < 0.0001 between RpfE- and RpfE+NS398-treated mice.

**Figure 4 ijms-22-07535-f004:**
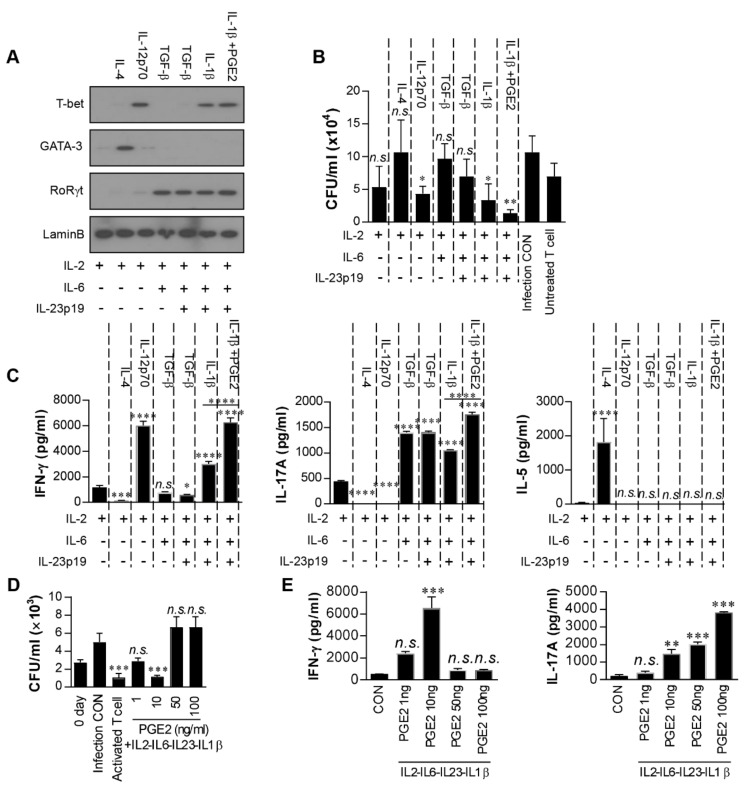
Low-dose prostaglandin E2 (PGE2) promotes Th1/Th17 differentiation in vitro. Naïve CD4^+^ T cells (1 × 10^6^) were cultured with cytokines (5 ng/mL IL-2; 25 ng/mL IL-4; 5 ng/mL IL-12; 25 ng/mL IL-6; 250 ng/mL TGF-β; 50 ng/mL IL-23; 10 ng/mL IL-1β; 5 ng/mL PGE2) in 24-well culture plates coated with anti-CD3 antibodies (2 μg/mL) for 5 days. (**A**) The levels of T-bet, GATA-3, and RoRγt in differentiated T cells induced by each combination of cytokines were assessed by immunoblotting. (**B**) Differentiated T cells induced by each combination of cytokines were co-cultured with *Mycobacterium tuberculosis* (Mtb)-infected bone marrow-derived macrophages (BMDMs). Intracellular Mtb growth in the BMDMs was determined after 3 days. All data are expressed as the mean ± SD; * *p* < 0.05 and ** *p* < 0.01 for treated samples compared to infection control; *n.s.*, no significant difference. (**C**) The quantities of IFN-γ, IL-17A, and IL-5 in the culture supernatants were determined using ELISA. (**D**,**E**) T cells were generated by stimulating naïve T cells with 5 ng/mL IL-2, 25 ng/mL IL-6, 250 ng/mL TGF-β, 50 ng/mL IL-23, 10 ng/mL IL-1β and PGE2 (1, 10, 50 or 100 ng/mL) for 96 h. All data are expressed as the mean ± SD; * *p* < 0.05, *** *p* < 0.001 and **** *p* < 0.0001 for treated samples compared to IL-2 treated T cell; *n.s.*, no significant difference. (**D**) These T cells were co-cultured with Mtb-infected BMDMs. Intracellular Mtb growth in the BMDMs was determined after 3 days. Activated T cell was positive control and T cells activated by RpfE-matured DCs. Data are the mean ± SD from three independent experiments; *** *p* < 0.001 compared to infection control. *n.s.*: no significant difference. (**E**) The culture supernatants were determined using ELISA. Data are the mean ± SD from three independent experiments; ** *p* < 0.01 and *** *p* < 0.001 vs. appropriate controls. *n.s.*: no significant difference.

## Data Availability

All data are contained within the manuscript.

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
