# Peer review of "Mycobacterium tuberculosis RpfE-Induced Prostaglandin E2 in Dendritic Cells Induces Th1/Th17 Cell Differentiation"

_ijms, 2021, doi:10.3390/ijms22147535_

Round 1

Reviewer 1 Report

This work represents continuation of the author’s long-lasting attempts to link DC activation, T-cell polarization and PGE2 production with biological activities of mycobacterial proteins belonging to the Rpf family. Here they concentrate on the approach based upon blockage of T-cell phenotypes with different low m. w. molecules and antibodies specific for PGE2 receptors/enzymes in the presence or absence of the RpfE protein. Being technically sound, their work predominantly focuses on the results obtained in cell co-culture systems, with marginal validation in vivo: multiple intranasal Rpf administrations via respiratory tract of B6 mice infected with non-virulent H37Ra mycobacteria have only weak influence onto mycobacterial survival. This infection model does not mimic progressive or chronic TB infections and thus is of limited value – 2-6-fold differences in CFU counts during persistence of non-virulent mycobacteria are not impressive from my point of view. Moderate differences between experimental groups concerning highly variable immunological phenotypes in cell cultures were significant, but their linkage with real physiology remains vague.  On the other hand, the experiments were performed accurately, the amount of data obtained is sufficient, the hierarchy of PGE2 receptors involved in the pathway has been identified, so the manuscript provides useful information and as such may be published.

     Before I summarize my particular concerns (see below), I should emphasize that the manuscript requires intensive editing, especially in its wording part, since quite few of sentences is difficult to understand. Some examples will illustrate this.

  There is no explanation how the authors have selected particular Rpf proteins for experimentation from the set of five. They worked with RpfB before, switched to RpfE here, but I found no general reasons for these choices in their previous publications and the present MS. A short paragraph on this subject would be a useful part of the Introduction.

   Line 374 – provide the reference in a proper way.

   Fig. 1A – mark Y-axis

   Lines 90-91 and 07-103 – “…RpfE-mediated PGE2 production was abrogated by pretreatment with a COX2 inhibitor (NS398) (Fig. 1C). However, the COX2 inhibitor did not inhibit production of RpfE-mediated Th1/Th17-related cytokines (Fig. 1D)…” – The message is understandable but the style interferes with clearness. I suggest rearranging the sentences to make the whole paragraph and conclusions clearer.

    Line 156 – “…BCG-infected mice splenocytes”? – Not a single word about BCG in the M&M section. Please explain.

   Line 174-175 - “Mtb-infected mice were intranasally vaccinated six times at two-day intervals with RpfE…” The verb “vaccinate” (as well as “immunize”) has nothing to do with the real aim of RpfE administration, the purpose of which was induction of PGE2-related shifts in phenotypes. Vaccination with a purified protein without adjuvants is immunologically ineffective, anyway. These terms should be replaced with more appropriate ones throughout the manuscript.

    Line 178 – “were significantly recovered”? If the loads were higher than in some other group it should be clearly stated. The authors have made no attempts to “recover” something.

    Line 185 – “completely reduced”: probably, completely abrogated?

There are many more examples of miswording, so please perform extensive editing.

Reviewer 2 Report

Manuscript Number: ijms-1263328

Mycobacterium Tuberculosis RpfE-Induced Prostaglandin E2 in Dendritic Cells Induces Th1/Th17 Cell Differentiation

 This article reports that RpfE-treated dendritic cells(DCs) effectively expanded the Th1 and Th17 cell responses simultaneously and PGE2 produced by RpfE-activated DCs via the MAPK and cyclooxygenase 2 signaling pathways (not JNK signaling) induces Th1 and Th17 cell responses mainly via EP4 receptor. Additionally, this research confirmed that PGE2-dependent induction of RpfE-specific Th1 and Th17 immune responses in vivo and in vitro both. Thus, RpfE-matured DCs produce PGE2 that induces Th1 and Th17 cell differentiation with potent anti-mycobacterial activity. The reports are well-designed, organized and descripted. But, several questions were occurred. I have minor questions.

  1. This article described that PGE2 produced by Mtb RpfE-matured DCs simultaneously induce both Th1 and Th17 cell polarization and the balance between Th1 and Th17 responses during infection is important for the development of efficacious vaccines against Mtb. Then, is it also contribute to balance between Th1 and Th17 response?
  2. This study revealed that COX-2 expression was significantly induced in macrophages with avirulent Mtb H37Ra but not H37Rv. Then, why do you think it didn't work for the H37Rv?
  3. This manuscript described that RpfE-induced DC produces the PGE2 and optimal PGE2 is essential for anti-mycobacterial effect. But, This study showed that COX2 is not expressed in H37Rv infection and it means PGE2 production is also decreased. It is better to add that how to regulate RpfE against pathogenic Mtb in discussion section.

[Fig1. Mycobacterium tuberculosis (M.tb) RpfE produces prostaglandin E2 (PGE2) in activated dendritic cells (DCs) via the MAPK/NF-B/COX2 signaling pathway.]

  1. There are many inhibitors in this paper. Others, the preceded by the object being inhibited, followed by the name of the inhibitor in parentheses. But, ‘COX-2’ is not followed this rule. I think it is good to unify this rule.

[Fig2. Prostaglandin E2 (PGE2) produced by RpfE-activated dendritic cells (DCs) induces Th1/Th17 differentiation via EP4 receptor.]

  1. This reports demonstrate, that EP4 is major receptor than EP2. But in figure 2E, the EP2 is more efficiency for anti-mycobacterial. So, how can we conclude that EP4 is more effective and main receptor?
  2. In figure 2, DC and splenocyte were co-culutred and treated RpfE. This means splenocytes were also exposed to RpfE. I wonder whether only treating the RpfE to splenocyte is related to differentiation for TH1/TH17.
  3. The data of Figure 3 suggest that RpfE-induced Th1 and Th17 responses are dependent on PGE2 in vivo. It is better to add the description why the COX2 inhibitor was chosen to prove the hypothesis in Figure 3.

[Fig4. Low-dose prostaglandin E2 (PGE2) promotes Th1/Th17 differentiation in vitro.]

  1. Optimal dose of PGE2 is essential to promote Th1/Th17 differentiation with strong bactericidal activity. Then, how to regulate the dose of PGE2 artificially if it is utilized to develop vaccine under nature conditions that express PGE2 constantly to defense against Mtb?
  2. In Figure 4A and 4B, synergetic effect of PGE2 for expressing transcription factors and inhibiting bacterial growth is only described in the condition of IL-2-IL-6-IL-23p19-IL-1β, even though it is also found in the condition of IL-2-IL-12p70. I wonder if there is any enhancement for Th1 differentiation when adding PGE2.
  3. In Figure 4C, the level of IL-17A was measured when concentration of used PGE2 is 5 ng/mL. However, IL-17 was produced in a PGE2 dose-dependent manner in Figure 4E. I wonder about measuring optimal IL-17 level required for T cell differentiation with strong bactericidal activity.
